# An automatic adaptive method to combine summary statistics in approximate Bayesian computation

**Jonathan U. Harrison**[1]*, **Ruth E. Baker**[2]

**1** Mathematical Institute, Mathematical Sciences Building, University of Warwick, Coventry, United Kingdom,
**2** Mathematical Institute, Andrew Wiles Building, University of Oxford, Oxford, United Kingdom

\* jonathan.u.harrison@warwick.ac.uk

## Abstract

To infer the parameters of mechanistic models with intractable likelihoods, techniques such as approximate Bayesian computation (ABC) are increasingly being adopted. One of the main disadvantages of ABC in practical situations, however, is that parameter inference must generally rely on summary statistics of the data. This is particularly the case for problems involving high-dimensional data, such as biological imaging experiments. However, some summary statistics contain more information about parameters of interest than others, and it is not always clear how to weight their contributions within the ABC framework. We address this problem by developing an automatic, adaptive algorithm that chooses weights for each summary statistic. Our algorithm aims to maximize the distance between the prior and the approximate posterior by automatically adapting the weights within the ABC distance function. Computationally, we use a nearest neighbour estimator of the distance between distributions. We justify the algorithm theoretically based on properties of the nearest neighbour distance estimator. To demonstrate the effectiveness of our algorithm, we apply it to a variety of test problems, including several stochastic models of biochemical reaction networks, and a spatial model of diffusion, and compare our results with existing algorithms.

## 1 Introduction

When using quantitative models to explore biological or physical phenomena, it is crucial to be able to estimate parameters of these models and account appropriately for uncertainty in both the parameters and model predictions. Bayesian statistics offers a wealth of tools in this regard [1, 2]. Bayes' theorem gives us that the posterior, $p(\theta|\mathbf{y})$, of parameters, $\theta$, given data, $\mathbf{y}$, is proportional to the prior, $\pi(\theta)$, on the parameters multiplied by the likelihood, $p(\mathbf{y}|\theta)$, of data, $\mathbf{y}$, given those parameters: $p(\theta|\mathbf{y}) \propto p(\mathbf{y}|\theta)p(\theta)$. The prior represents our beliefs about the parameters prior to observing the data, the likelihood gives the probability of observing the data, given a certain set of parameters, and these result in the posterior, which returns updated beliefs about the parameters after having observed the data.

**Data Availability Statement:** All relevant data are within the manuscript and its Supporting Information files.

**Funding:** This work was supported by funding from the Engineering and Physical Sciences

Research Council (EPSRC) (grant no. EP/
G03706X/1). Ruth E Baker is a Royal Society
Wolfson Research Merit Award holder and a
Leverhulme Research Fellow, and would also like
to thank the BBSRC for funding via grant number
BB/R00816/1. The funders had no role in study
design, data collection and analysis, decision to
publish, or preparation of the manuscript.

**Competing interests:** The authors have declared
that no competing interests exist.

However, much of the current theory surrounding the generation of posterior distributions for parameter inference relies on being able to evaluate the likelihood of the data given the parameters of a model. In practice, for a large class of mechanistic models the likelihood is not tractable, either due to computational or analytical complexity. Therefore, the use of likelihood-free methods for inference has become widespread [3]. These likelihood-free methods include indirect inference [4], synthetic likelihoods [5, 6], and expectation propagation [7]. Markov Chain Monte Carlo (MCMC) methods for doubly intractable distributions with parameter dependency in the normalizing constant have been developed [8, 9] using auxiliary variable methods, and these provide exact inference. Multiple Try Metropolis methods [10, 11] can improve mixing properties by comparing multiple reference draws at once and these methods offer an implicit approximation of the marginal likelihood. Similarly, pseudo-marginal MCMC and particle Markov Chain Monte Carlo (pMCMC) methods [12–16] provide exact inference for intractable distributions including state space models. However, pMCMC methods often have extremely low acceptance rates and poor mixing properties. Methods to mitigate these issues include the use of noisy Monte Carlo methods [17, 18], but these no longer provide exact inference, and correlating the pseudorandom numbers used to estimate the likelihood [19]. The focus of this work, however, will be on one of the most popular methods for likelihood-free inference, approximate Bayesian computation (ABC) [20–25], which has been widely adopted due to its ease of understanding and implementation.

## 1.1 Approximate Bayesian computation

Suppose we wish to infer a posterior distribution over parameters $\theta$ of a generative model such that we can simulate from $\mathbf{x} \sim f(\mathbf{x}|\theta)$. In ABC, parameters $\theta$ are drawn from a prior, $\pi(\theta)$, and data, $\mathbf{x}^*$, is simulated from the generative model using those parameters, such that $\mathbf{x}^* \sim f(\mathbf{x}|\theta)$. The distance between the simulated dataset, $\mathbf{x}^*$, and the real data, $\mathbf{y}$, is calculated using a distance function $d(\mathbf{x}^*, \mathbf{y})$. If this distance is less than a certain tolerance, $\epsilon$, then the parameters $\theta$ can be accepted into the approximate posterior sample. Choice of the tolerance $\epsilon$ can be avoided, to some extent, by simulating a large number, $N$, of parameter samples and datasets, calculating the corresponding distances for these and accepting the proportion $\alpha$ that lie closest to the real data [26, 27]. We will use this approach in this work.

ABC can be viewed as providing a regular Bayesian analysis, but with an approximation to the likelihood function of the form [28]:

$$p_{\text{ABC}}(\mathbf{y}|\theta) = \int 1[d(\mathbf{x}, \mathbf{y}) \leq \epsilon]p(\mathbf{y}|\theta)d\mathbf{x}, \tag{1}$$

where $p_{\text{ABC}}(\mathbf{y}|\theta)$ is the ABC approximation of the likelihood, and $p(\mathbf{y}|\theta)$ is the exact likelihood. The quality of the approximation depends on the choice of tolerance, $\epsilon$, and distance function $d(\mathbf{x}, \mathbf{y})$.

In cases where the prior and posterior distributions are very different, the rejection sampling version of ABC described above can have very low acceptance rates. Algorithm 1 summarizes how samples from an approximate posterior can be generated via a more efficient version of ABC using sequential Monte Carlo (SMC) techniques, known as ABC-SMC [29–31]. Importance sampling is used iteratively so that instead of sampling repeatedly from the prior, parameters are sampled from an approximate posterior at each generation of the algorithm. A weight must be given to each sample to correct for the fact that it is not drawn from the prior.

**Algorithm 1** ABC-SMC

```
1: for t = 1 to T do
2:    Set i = 1.
```

```
 3:   while i ≤ N do
 4:     if t==1 then
 5:        Sample from prior θ₁ⁱ ~ π(θ).
 6:        Set weight to ν₁ⁱ = 1/N. Set i = i + 1.
 7:     else
 8:        Sample θ* from previous population {θᵢ₋₁ⁱ}ᴺᵢ₌₁ with weights {νₜ₋₁}ᴺᵢ₌₁.
           Perturb θ* to give θ** ~ Kₜ(θ|θ*). If π(θ**) = 0, then resample
           θ* and repeat.
 9:        Simulate dataset x*ⁱ ~ f(x|θ**) and calculate distance d(s
           (x*ⁱ), s(y)).
10:        if d(s(x*ⁱ), s(y)) < εₜ then
11:           Accept the parameters.
              Set θₜⁱ = θ**.
              Calculate the weight νₜⁱ for particle θₜⁱ via
```

$$\nu_t^i = \frac{\pi(\theta_t^i)}{\sum_{j=1}^{N} \nu_{t-1}^j K_t(\theta_{t-1}^j, \theta_t^i)}.$$

```
12:           Set i = i + 1.
13:        end if
14:     end if
15:   end while
16:   Normalize the particle weights {νₜⁱ}ᴺᵢ₌₁.
17: end for
18: return {θ_Tⁱ}ᴺᵢ₌₁, {ν_Tⁱ}ᴺᵢ₌₁
```

## Regression adjustments

Post-processing methods to aid with some of the approximations of ABC have long been advocated [21]. Regression adjustments can correct for the relationship between parameters, θ, and summary statistics, s(x), in the sense of accounting for the imperfect match between observed and simulated data [20]. Further extensions include non-linear, heteroscedastic regression adjustment [32] and ridge regression approaches [33].

## 1.2 Role of summary statistics in ABC

Suppose we are interested in inferring multi-dimensional parameters for a model that we can simulate, but cannot evaluate the likelihood directly. In many practical circumstances, the data (either collected experimentally or simulated from the *in silico* model) will be very high dimensional. Such high-dimensional data poses difficulties within the ABC framework, as it is difficult to sensibly estimate when the output of a particular simulation is 'close' to the data. Even taking account of domain expertise, it can be hard to determine which features of the data are important. This issue of comparing high-dimensional data is further compounded using stochastic models where there is noise in the process model in addition to measurement noise. Repeatedly drawing from a stochastic model with the same parameter values can give vastly different outputs.

As such, it is often necessary to work with a lower-dimensional vector of summary statistics, $s(\mathbf{x})$, of the data, such that we require the distance between summary statistics is less than the tolerance, $d(s(\mathbf{x}^*), s(\mathbf{y})) < \epsilon$. Examples of these summary statistics may be data points within a time series, an average transition time between different states of a system, or the moments of a certain species within a model. However, not all summary statistics are equally informative about the posterior. Common practice is to combine summary statistics based on some

heuristic approach, such as weighting the contribution of each summary statistic according to its standard deviation. However, it is not clear whether these heuristic approaches result in optimal weighting of the various summary statistics available. As such, the aim of this work is to provide an automated and adaptive method for determining the weighting of available summary statistics in order to optimize the quality of the resulting posterior.

Previous work has also considered how to weight or select summary statistics for ABC. Fearnhead and Prangle [34] developed a popular method to find informative linear combinations of summary statistics by fitting a regression for each model parameter. The result is a reduction from the original high-dimensional set of summary statistics to a new lower dimensional set of summary statistics with the same dimensionality as the parameter space. Improved results are seen by using a pilot run of ABC to choose a subset of parameter space as a training region for the regression. Further improvements are obtained by extending the vector of summary statistics by concatenating with a non-linear transformation of the same summary statistics, $s(\mathbf{x}) = (s, s^2, s^3, s^4)$, where $s$ is a given vector of summary statistics and the superscripts indicating raising these to the given power. This method uses contributions from all of the summary statistics and should optimize the mean quadratic loss.

Other successful approaches rely on subset selection methods [35–37]. Joyce and Marjoram [36] use criteria for approximate sufficiency of a subset of statistics to test whether adding a new statistic results in a change in the posterior above a certain threshold. [37] minimize an information criterion based on $k$ nearest neighbour entropy over all subsets of summary statistics. Barnes et al. [35] use an approximate sufficiency criterion to select a subset of summary statistics on which to base inference, by adding summary statistics until the Kullback-Liebler (KL) divergence between the resulting posteriors is below a threshold in order to reduce any loss of information. All of these methods seek to choose a lower dimensional subset of a given list of summary statistics. Using this lower dimensional subset increases the acceptance rate for samples in ABC by avoiding the curse of dimensionality for the data. However, the results can depend on the order in which the summary statistics (or subsets) are analysed.

Further methods rely on assigning weights to each summary statistic within the ABC distance function and considering how best to choosing these weights. Prangle [38] present a method for adaptively choosing summary statistic weights for ABC based on the scale of the summary statistics from the predictive distribution for summary statistics at each generation of an ABC-SMC algorithm. The median absolute deviation (MAD), a measure of spread of a statistic, is used for the scaling. All summary statistics are scaled by a measure of their spread, but if the summary statistics are not equally informative about the posterior distribution this may not be the best choice. A genetic algorithm has been used to choose the weights of different summary statistics [39]. This genetic algorithm attempts to optimize the mean squared error (MSE) of the posterior samples from the true parameter. Recently, Singh and Hellander [40] have proposed a multi-armed bandit problem approach to selecting summary statistics for ABC. Other work has avoided using summary statistics at all by considering the Wasserstein distance between full data sets [41, 42] or by using classification techniques from machine learning to discriminate between datasets [43], and thus choosing a classification method and features rather than a distance function and summary statistics.

In this work, we approach the problem from the point of view of finding the optimal distance function, adapted to information contained in a fixed list of known summary statistics, rather than selecting a certain subset of summary statistics. Following the perspective of ABC offering an approximation to the likelihood, as in Eq (1), we can consider how optimizing the choice of ABC distance function, given a set of summary statistics, can improve the approximation to the likelihood offered by ABC. We provide an automatic algorithm that adaptively selects weights for each summary statistic within the ABC distance function.

### 1.3 Outline

Our contribution in this work is to present a flexible, novel framework for improving inference with ABC by adapting the weights of different summary statistics to maximize the gain in posterior information from a dataset. This is helpful for avoiding bias and variance from redundant information in data (such as would be the case when including a summary statistic that is uncorrelated with the parameters of interest). A further advantage of our work is that it can alleviate the burden of designing and selecting summary statistics 'by hand', since a large collection of summary statistics can be used and weighted appropriately via our procedure. It is also possible to combine our framework with existing dimensionality reduction techniques for summary statistics in ABC (see Section 4.7).

We outline in Section 2 our adaptive algorithm for combining summary statistics in an ABC framework. We provide theoretical justification for the algorithm in Section 3 and demonstrate that we obtain convergence to the posterior distribution. To demonstrate the utility of our algorithm, we apply it to several test problems based on biochemical reaction networks in Section 4. We compare results of parameter inference using our algorithm against benchmark results from applying ABC-SMC using other choices of weights for the summary statistics. In Section 4.5, we evaluate the additional computational cost of our method versus the benefits it offers for parameter inference. Finally, in Section 5, we summarize the work presented in this article and compare our methodology for combining summary statistics with other techniques in the literature that are based on dimensionality reduction of a set of summary statistics.

## 2 An algorithm for automatic weighting of summary statistics

A SMC algorithm produces a sequence of approximations that aid in moving between the prior and posterior distributions. The motivation for the method introduced here is to accelerate that movement from the prior to the posterior by maximizing the distance between the prior and the current approximation, subject to the procedure being a valid ABC-SMC algorithm.

In order to use ABC-SMC (see Algorithm 1), we must specify a function to measure the distance between simulated and real datasets. Suppose we take a weighted Euclidean distance as the ABC distance function such that

$$d_{\mathbf{w}}(s(\mathbf{x_1}), s(\mathbf{x_2})) = \sum_{i=1}^{\kappa} w_i (s_{1i} - s_{2i})^2,$$

where $s(\mathbf{x}) = (s_1, \ldots, s_{\kappa}) \in \mathbb{R}^{\kappa}$ is a vector of summary statistics and the sum over $i$ is taken over all the summary statistics considered. This distance function is a reasonable and flexible choice commonly used in the literature [44]. It is these distance weights, $w_i$, that control how the summary statistics are combined in this case. Given simulated pairs of parameter samples and datasets, we find weights, $\mathbf{w} = (w_1, \ldots, w_{\kappa}) \in E \subset \mathbb{R}^{\kappa}$, that maximize a distance between the prior and the posterior that represents the maximum possible gain in information about the parameters from the given data. Constructing the weights in this way allows us to account for the scale of the summary statistics, as well as their relative contribution to a posterior.

### 2.1 Adaption of weights

We seek to optimize the weights, $\mathbf{w}$, so that we can place less emphasis on summary statistics that are not informative for the posterior, but also scale summary statistics appropriately so that we do not neglect to obtain information about certain parameters. We do this within the

ABC-SMC framework [29–31] given in Algorithm 1 using the implementation outlined in [38]. We outline our proposed methodology in Algorithm 2.

At each generation, we search for the weights, **w**, of the distance function that maximize the distance between the prior and resulting posterior, given $N$ ABC samples from the model for different $\theta$ values. This distance between prior and posterior gives a measure of the information gain in moving from the prior to the posterior.

## 2.2 Distance between distributions

We use the Hellinger distance to measure the discrepency between the prior and approximate posterior, and so to quantify the information gained in moving towards the true posterior distribution. The Hellinger distance is defined, for distributions $P$ and $Q$, with densities $p$ and $q$, respectively, as

$$H^2(P, Q) \quad = \frac{1}{2} \int \left( \sqrt{p(x)} - \sqrt{q(x)} \right)^2 \mathrm{d}x$$

$$= 1 - \int \sqrt{p(x)q(x)} \, \mathrm{d}x.$$

Alternative measures of distance between distributions such as the Euclidean distance or KL divergence can be used. In our experience, the Hellinger distance performs better than alternatives, particularly for robustly identifying relatively small differences between posterior distributions when weights are optimized, an observation that is supported by other work [45]. In addition, the Hellinger distance is finite when comparing distributions with different support (unlike the KL divergence). This property is desirable when comparing a broad prior with a posterior distribution where we have gained some knowledge of parameter space and can exclude certain regions.

### Nearest neighbour distance estimator

To estimate the distance between two distributions, based on samples from these distributions, we use a $k$ nearest neighbour estimator [46–48] developed to describe a family of distances between distributions known as $\alpha$ divergences, of which the Hellinger distance is a special case. Suppose we have two probability distributions, $P$ and $Q$ with densities $p(x)$, and $q(x)$, and are interested in the distance between these. We suppose that we have some samples, $X_{1:N}$ and $Y_{1:N}$ from $p$ and $q$. If we define $D_\alpha(p\|q) = \int p^\alpha(x)q^{1-\alpha}(x)\mathrm{d}x$ for $\alpha \in \mathbb{R}$, then the Hellinger distance, which is related to the divergence $D_\alpha$ for $\alpha = 1/2$, is defined as

$$D_h(p\|q) = 1 - D_{1/2}(p\|q).$$

The $k$ nearest neighbour estimator that we use depends only on distances between observations in a sample. Let $\rho_k(i)$ be the Euclidean distance from the sample $X_i$ to its $k$th nearest neighbour in $X_{1:N}$. Similarly, let $\nu_k(i)$ be the distance from $X_i$ to its $k$th nearest neighbour in the samples $Y_{1:N}$. Then the estimator [46] is given by

$$\hat{D}_\alpha(X_{1:N}\|Y_{1:N}) = \frac{1}{N} \sum_{i=1}^{N} \left( \frac{(N-1)\rho_k(i)}{N\nu_k(i)} \right)^{1-\alpha} B_{k,\alpha}, \tag{2}$$

where

$$B_{k,\alpha} = \frac{\Gamma(k)^2}{\Gamma(k-\alpha+1)\Gamma(k+\alpha-1)}.$$

At each generation of ABC-SMC we seek to find weights $\mathbf{w} \in E \subset \mathbb{R}^{\kappa}$ such that

$$\mathbf{w}^* = \underset{\mathbf{w} \in E}{\operatorname{argmax}}(1 - \hat{D}_{\alpha}(\{\xi_i\}_{i=1}^N \| \{\theta_i\}_{i=1}^N)), \tag{3}$$

where $\{\xi_i\}_{i=1}^N$ are samples from the prior distribution, and $\{\theta_i\}_{i=1}^N$ are samples from the approximate posterior distribution, which depends on the summary statistic weights, $\mathbf{w}$.

To perform the optimization in weight space in our implementation of Algorithm 2, we use a constrained nonlinear optimizer, implemented via `fmincon` in MATLAB [49], with multiple initializations of the optimizer. This nearest neighbour estimator is computationally intensive in high dimensions, where the high-dimensional setting corresponds to using many particles in SMC part of the adaptive ABC-SMC algorithm. In the context of ABC, the computational cost is dominated nonetheless by the expensive model simulations.

**Algorithm 2** Adaptive ABC-SMC

```
1:  Set generation index t = 1. Let M = ⌈N/α⌉.
2:  Set i = 1.
3:  while i ≤ M do
4:      Sample from prior θ** ∼ π(θ).
5:      Simulate dataset x*ⁱ ∼ f(x|θ**).
6:      Accept the parameters. Set i = i + 1.
        Set θᵢₜ = θ**.
        Calculate the weight vᵢₜ for particle θᵢₜ as vᵢₜ = 1.
7:  end while
8:  Let
```

$$L(\mathbf{w}) = 1 - \hat{D}_{1/2}(\{\xi^i\}_{i=1}^N \| \{\theta_t^i\}_{i=1}^N),$$

```
    where {θᵢₜ}ᴺᵢ₌₁ are the closest N samples when ranked according to ABC dis-
    tance from the observed dataset, y, via d�w(s(x^{i*}), s(y))).
9:  Maximize L(w) as a function of summary statistic weights, w.
10: Keep the samples {θᵢₜ}ᴺᵢ₌₁ corresponding to the maximum of L(w) (i.e.
    the maximum distance between prior and approximate posterior).
11: Normalize the particle weights {vᵢₜ}ᴺᵢ₌₁.
12: Calculate tolerance, ε_t, as the α quantile of distances. Store the
    optimum weights w*ₜ.
13: for t = 2 to T do
14:     Set i = 1.
15:     while i ≤ M do
16:         Sample θ* from previous population {θᵢₜ₋₁}ᴺᵢ₌₁ with weights {vᵢₜ₋₁}ᴺᵢ₌₁.
17:         Perturb θ* to give θ** ∼ Kₜ(θ|θ*). If π(θ**) = 0, resample θ* and
            repeat.
18:         Simulate dataset x*ⁱ ∼ f(x|θ**).
19:         if d_{w*ⱼ}(s(x^{i*}), s(y)) < ε_j for all j < t then
20:             Accept the parameters.
                Set θᵢₜ = θ**.
```

Calculate the weight $v_t^i$ for particle $\theta_t^i$ as

$$v_t^i = \frac{\pi(\theta_t^i)}{\sum_{j=1}^{M} v_{t-1}^j K_t(\theta_{t-1}^j, \theta_t^i)}.$$

21:     **end if**
22:     Set $i = i + 1$.
23:  **end while**
24:  Let

$$L(\mathbf{w}) = 1 - \hat{D}_{1/2}(\{\xi^i\}_{i=1}^N \| \{\theta_t^i\}_{i=1}^N),$$

where $\{\theta_t^i\}_{i=1}^N$ are the closest $N$ samples when ranked according to ABC distance from the observed dataset, $\mathbf{y}$, via $d_{\mathbf{w}}(s(\mathbf{x}^{i^*}), s(\mathbf{y}))$.
25:  Maximize $L(\mathbf{w})$ as a function of summary statistic weights, $\mathbf{w}$.
26:  Keep the samples $\{\theta_t^i\}_{i=1}^N$ corresponding to the maximimum of $L(\mathbf{w})$ (i.e. the maximum distance between prior and approximate posterior).
27:  Normalize the particle weights $\{v_t^i\}_{i=1}^N$.
28:  Calculate tolerance, $\epsilon_t$, as the $\alpha$ quantile of distances. Store the optimum weights $\mathbf{w}_t^*$.
29: **end for**
30: **return** $\{\theta_T^i\}_{i=1}^N$, $\{v_T^i\}_{i=1}^N$

## 3 Theoretical justification

In the following, we demonstrate that by using the nearest neighbour estimator, we can obtain optimum weights that maximize the distance between the prior and approximate posterior. Furthermore, we show by using standard arguments that the adaptive ABC-SMC algorithm given by Algorithm 2 will converge to the correct ABC posterior distribution.

### 3.1 Convergence of the nearest neighbour estimator

The estimator we use is a $k$ nearest neighbour estimator relying only on distances between observations in a sample, as described above in Section 2.2. With this nearest neighbour estimator, Eq (2), under conditions on $p$ and $q$, we have $L_2$ convergence of the estimator [46] and this ensures that, in the limit of more samples, estimates of the distance between $p$ and $q$ will become more concentrated around the true distance, such that optimizing the estimate of the distance will give the true optimum, $\mathbf{w}^*$. We note that although the Hellinger distance, $D_h(p\|q)$, is symmetric in $p$ and $q$, the estimator above in Eq (2) is not. By using the estimator from Eq (2) and choosing $q$ as the distribution that depends on the weights, $\mathbf{w}$, we are able to make strong assumptions about $p$ independent of the weights, $\mathbf{w}$, and make weaker assumptions about $q$. In the context of our algorithm for ABC, this allows us to treat $p$ as the prior and $q$ as the approximate posterior distribution.

   **Theorem 1**. *Suppose that $k \geq 2$ and that $\mathcal{M} = supp(p)$. Assume that*

(a).   *$q$ is bounded above,*

(b).   *$p$ is bounded away from zero,*

(c).   *$p$ is uniformly Lebesgue approximable,*

(d).  $\exists \delta_0$ such that $\forall \delta \in (0, \delta_0)$

$$\int_{\mathcal{M}} H(x, p, \delta, 1/2) p(x) \mathrm{d}x < \infty,$$

where $H(x, p, \delta, \psi) =$

$$\sum_{j=0}^{k-1} \left(\frac{1}{j!}\right)^{\psi} \Gamma\left(\frac{1}{2} + j\psi\right) \left(\frac{p(x) + \delta}{p(x) - \delta}\right)^{j\psi} (p(x) - \delta)^{-\frac{1}{2}}$$

$$((1 - \delta)\psi)^{-\frac{1}{2} - j\psi}.$$

(e).  $\int_{\mathcal{M}} \|x - y\|^{\gamma} p(y) \mathrm{d}y < \infty$ for almost all $x \in \mathcal{M}$, $\int \int_{\mathcal{M}^2} \|x - y\|^{\gamma} p(y) p(x) \mathrm{d}y \mathrm{d}x < \infty$,

(f).  $E \subset \mathbb{R}^{\kappa}$ is compact,

(g).  a unique $\mathbf{w}^* \in E$ maximizes $D_h(p\|q^{(\mathbf{w})})$.
     Then

$$\lim_{N \to \infty} \mathbb{P}\left(\underset{w \in E}{\mathrm{argmax}}\, \hat{D}_h(X_{1:N}\|Y_{1:N}^{(\mathbf{w})}) = \mathbf{w}^*\right) = 1. \tag{4}$$

*Proof.* See S1 File for details.

The proof of Theorem 1 relies on results from [46] showing $L_2$ consistency of the nearest neighbour estimator, which ensures that the estimates of the distance between $p$ and $q$ become more concentrated around the true values as more samples are used. This requires the construction of an integrable function as a bound such that Lebesgue's dominated convergence theorem can be applied. Most of the conditions on $p$ and $q$ are to ensure that this all holds. We assume that the space of ABC distance weights, $E$, is compact. In practice this is not a problem, since we can work with a constrained optimization problem and assume that the summary statistic weights lie within a large but finite region. Although there are several conditions on the prior distribution, $p$, most reasonable choices of prior distribution will satisfy these, and only a single condition on the approximate posterior distribution, $q$, is assumed. In the limit of having more samples from the distributions $p$ and $q^{(\mathbf{w})}$, selecting summary statistic weights, $\mathbf{w}$, based on optimizing the estimate from $\hat{D}_h$ will converge to give the true optimum, $\mathbf{w}^*$, of this distance between distributions.

## 3.2 Convergence of adaptive ABC-SMC

We demonstrate here that Algorithm 2 converges to the correct target distribution by using arguments from [38].

**Theorem 2** [38]

*Suppose that the following conditions hold*:

(a).  $\theta \in \mathbb{R}^{\lambda}$ $s(\mathbf{x}) \in E \subset \mathbb{R}^{\kappa}$ and these random variables have joint density $\pi(\theta, s(\mathbf{x}))$ with respect to the Lebesgue measure;

(b).  the sets $A_t = \{s(\mathbf{x}) \mid d_{\mathbf{w}_t^*}(s(\mathbf{x}), s(\mathbf{y})) < \epsilon_t\}$ are Lebesgue measureable, where $\mathbf{w}_t^*$ are the optimal weights at generation t, and $\epsilon_t$ are the tolerances at each generation;

(c).  $\pi(s(\mathbf{y})) > 0$;

(d).  $\lim_{t \to \infty} m(A_t) = 0$, *where $m(\cdot)$ represents Lebesgue measure;*

(e).  *the sets $A_t$ have bounded eccentricity. That is for any $A_t$, $\exists$ a ball $B_t = \{s(\mathbf{x})\,|\,\|s(\mathbf{x} - s(\mathbf{y})\|_2 \leq r_t\}$ about $s(\mathbf{y})$ with radius $r_t$ such that $A_t \subset B_t$ and $m(A_t) \geq cm(B_t)$, where $\|\cdot\|$ is the Euclidean or $L_2$ norm and $c > 0$ is a constant.*
*Then the posterior approximations converge:*

$$\lim_{t \to \infty} p_{\text{ABC},t}(\theta|s(\mathbf{y})) = p(\theta|s(\mathbf{y})) \quad \text{for almost all } (\theta, s(\mathbf{y})),$$

*where $p_{\text{ABC},t}(\theta|s(\mathbf{y}))$ is the ABC posterior defined as*

$$p_{\text{ABC},t}(\theta|s(\mathbf{y})) \propto \int p(s|\theta)\pi(\theta)1[d_{\mathbf{w}_t^*}(s(\mathbf{x}), s(\mathbf{y})) \leq \epsilon_t]\,d\mathbf{x}.$$

*Proof.* See S1 File for details.

Similarly to [38], we can take $A_t = \{s(\mathbf{x}) \,|\, d_{\mathbf{w}_t^*}(s(\mathbf{x}), s(\mathbf{y})) < \epsilon_t \text{ for all } i \leq t\}$ as the acceptance region for Algorithm 2, which ensures that $m(A_t)$ is decreasing in $t$. This acceptance region is not guaranteed to converge to zero under all circumstances due to model misspecification or particles becoming trapped in the wrong mode of a multimodal posterior within the SMeC algorithm. However, these are issues that equally affect the conventional ABC-SMC algorithm, as discussed in [38]. The bounded eccentricity condition is ensured for the scaled Euclidean distance in Algorithm 2 by using weights $\mathbf{w} \in E$ with compact support, since the weights will be bounded by a maximum upper bound at each generation. Thus the conditions for convergence of Algorithm 2 are essentially the same as those for convergence of standard ABC-SMC algorithms, provided that the weights considered in the distance function have compact support.

## 4 Examples

We apply the adaptive ABC-SMC algorithm for weighting of summary statistics to a variety of test problems, including toy models and problems based on different chemical reaction networks. The dynamics of these networks are simulated stochastically using Gillespie's direct method [50], which allows trajectories to be sampled directly from the model. The summary statistics collected for each of the chemical reaction network problems are in the form of a time series, to imitate data that could be collected from a biological experiment. For some of these models it is possible to solve for the likelihood analytically, and we show comparisons with exact posterior distributions sampled via MCMC. However, for some of the more complex, but more biologically realistic, test problems, the likelihood is very computationally expensive to compute and the exact posterior distribution is not available.

To demonstrate the effectiveness of taking a flexible choice of distance weights, we make two comparisons. Firstly, we compare results obtained using Algorithm 2 to those generated using a uniform choice of weights: $w_i = 1 \, \forall i$. Secondly, we compare to results generated using weights that scale with each summary statistic, as in the method of [38]. For this method, we use $w_i = 1/\sigma_i \, \forall i$, where $\sigma_i$ is the MAD of the given summary statistic based on simulations from a given generation of ABC-SMC. We show approximate posterior distributions produced by each method for choosing weights. For the tractable test problems considered, we compute the MSE, Hellinger distance between prior and approximate posterior, and Hellinger distance between approximate posterior and exact posterior sampled via MCMC, all averaged over 40 repeats of Algorithm 2.

We note that a table summarising the parameters used in the implementation of all the test problems can be found in S1 File. For the $k$ nearest neighbour estimator, $k = 5$ is used throughout, as recommended by previous work [47].

## 4.1 Uniform toy model

We consider a tractable toy problem with a sufficient statistic to illustrate our method. We observe

$$x_i \sim \text{Unif}([0, \theta]), \tag{5}$$

for $i = 1, \ldots, r$. In this case, the maximum of the observed values is a sufficient statistic, $s(\mathbf{x}) = \max_i x_i$. We input into Algorithm 2 the full vector of data points sorted in ascending order, such that the summary statistics are the order statistics of the sample $s(\mathbf{x}) = (x_{(1)}, \ldots, x_{(r)})$. We can sample the true posterior distribution $p(\theta|s(\mathbf{y}))$ directly via (likelihood-based) MCMC and compare to the approximations obtained via ABC.

Here in Fig 1 we show marginal posterior distributions generated using uniform weights, weights scaled with the MAD of each summary statistic via the adaptive method of Prangle [38], and adaptively chosen weights via the method outlined in Algorithm 2. The results indicate that the method in Algorithm 2 is able to produce a higher quality approximation of the posterior for a given number of parameter samples compared to other methods of weighting the summary statistics. The true parameter is $\theta = 10$, a prior uniform on the logarithm of the parameters over the interval $[10^0, 10^2]$ was used, and $r = 10$ samples of the uniform model were used as the dataset.

## 4.2 Bimodal model

Although distances between distributions can be intuitive for unimodal Gasussian-like distributions, we want to ensure our method is robust to multimodal posterior distributions. The benefit of using a general distance measure like the Hellinger distance is that it is well suited to quantifying distance between such general distributions. In this second example, we aim to infer $\theta = (\theta_1, \theta_2) \in \mathbb{R}^2$ based on observations

$$s(\mathbf{x}) \sim N\left( \begin{pmatrix} \sin(\theta_1) \\ \sin(\theta_2) \end{pmatrix}, 0.1^2 I_2 \right).$$

We take a prior distribution for $\theta_1$ and $\theta_2$ uniform on $[0, 2\pi]$. We suppose that we observe $s(\mathbf{y}) = (\sqrt{2}/2, -\sqrt{2}/2)$, which leads to four distinct modes of a multimodal posterior distribution in two dimensions. As shown in Fig 2, the adaptive method is robust to multimodal posterior distributions and is competitive with the other methods considered. There are added complications in evaluating the quality of an approximation to a multimodal posterior distribution, since several distinct regions of parameter space can equally well explain the observed data. Using the symmetry of the posterior, we measure MSE by distance of the approximate posterior mean from $(\pi/2, \pi/2)$ at the centre of the four modes.

## 4.3 Dimerization system

To examine a system with multiple scales, we consider also a dimerization system, which undergoes a fast initial transient followed by slower subsequent dynamics [51]. This system provides a more biologically realistic example with higher dimensional parameter and

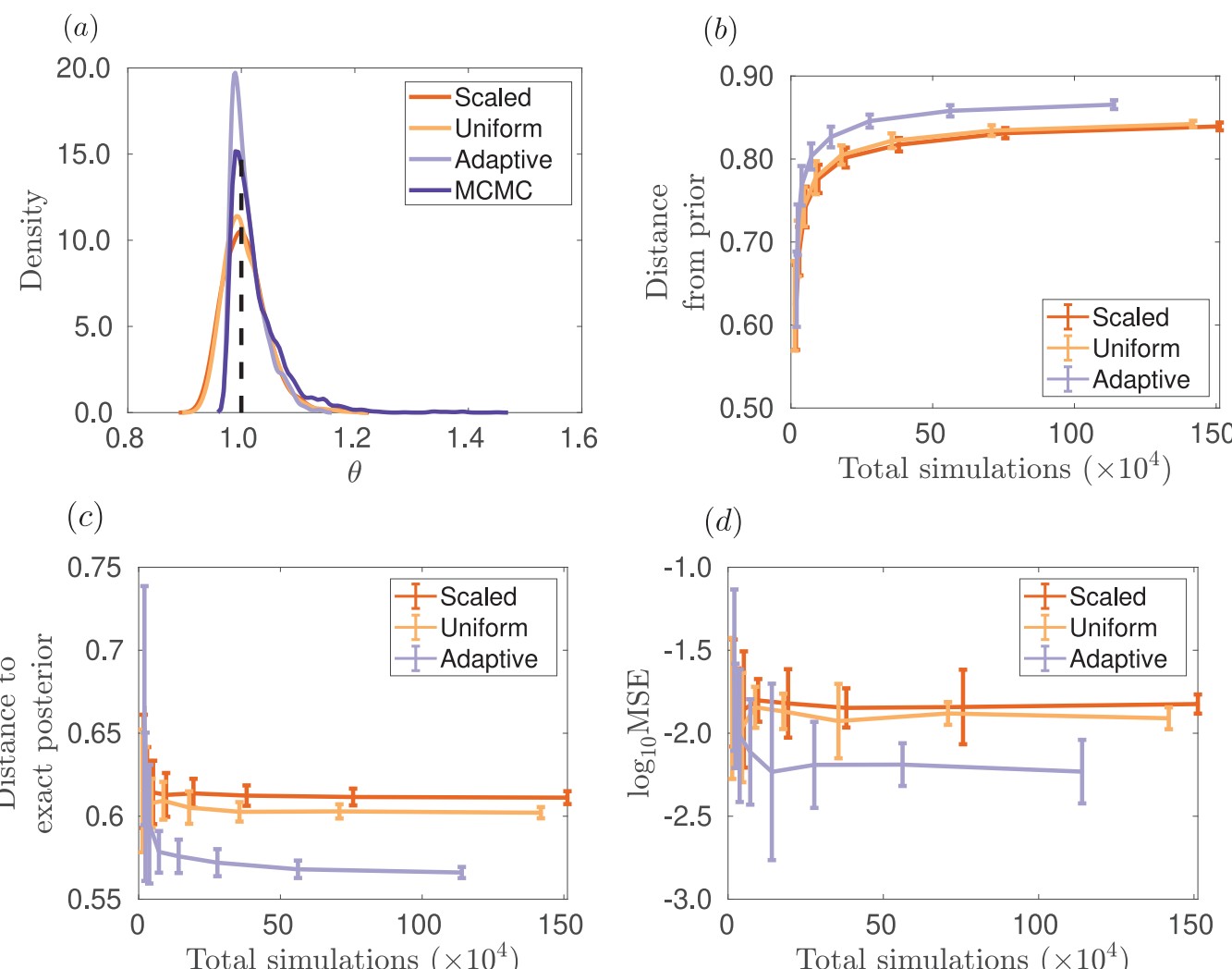

**Fig 1. Posterior for parameter $\theta$ of the uniform toy model for different weights in the ABC distance function.** The posterior distribution for parameter $\theta$ of the uniform toy model for different weights in the ABC distance function is shown in (a). ABC-SMC was used to provide estimates of the posterior, with 10 generations and $N = 2,000$ particles at each generation with the posterior constructed from the closest 50% of the simulations ($\alpha = 0.5$). Metrics to evaluate the performance of Algorithm 2 are shown in (b), (c), and (d) as $N$ varies resulting in different total numbers of simulations from the model. Results are averaged over 40 repeated runs. In (b), is shown the Hellinger distance from the prior to approximate posterior distribution, which is maximized directly within Algorithm 2. In (c), the Hellinger distance from the approximate posterior to the exact posterior distribution (as sampled via MCMC) is displayed. A lower value of this distance indicates a better approximation to the posterior. In (d), the MSE from the approximate posterior mean to the exact posterior mean is shown. The orange, yellow and light purple lines show the scaled, uniform and adaptive methods, respectively. For comparison as a gold standard for this problem, the dark purple line shows the posterior obtained with MCMC using the exact likelihood without any ABC approximation. The true parameter value used to simulate the observed data is indicated by the vertical dashed black line.

summary statistic spaces. The dimerization system consists of the following reactions:

$$R_1 \quad : \quad S_1 \xrightarrow{k_1} \emptyset;$$

$$R_2 \quad : \quad S_2 \xrightarrow{k_2} S_3;$$

$$R_3 \quad : \quad S_1 + S_1 \xrightarrow{k_3} S_2;$$

$$R_4 \quad : \quad S_2 \xrightarrow{k_4} S_1 + S_1.$$

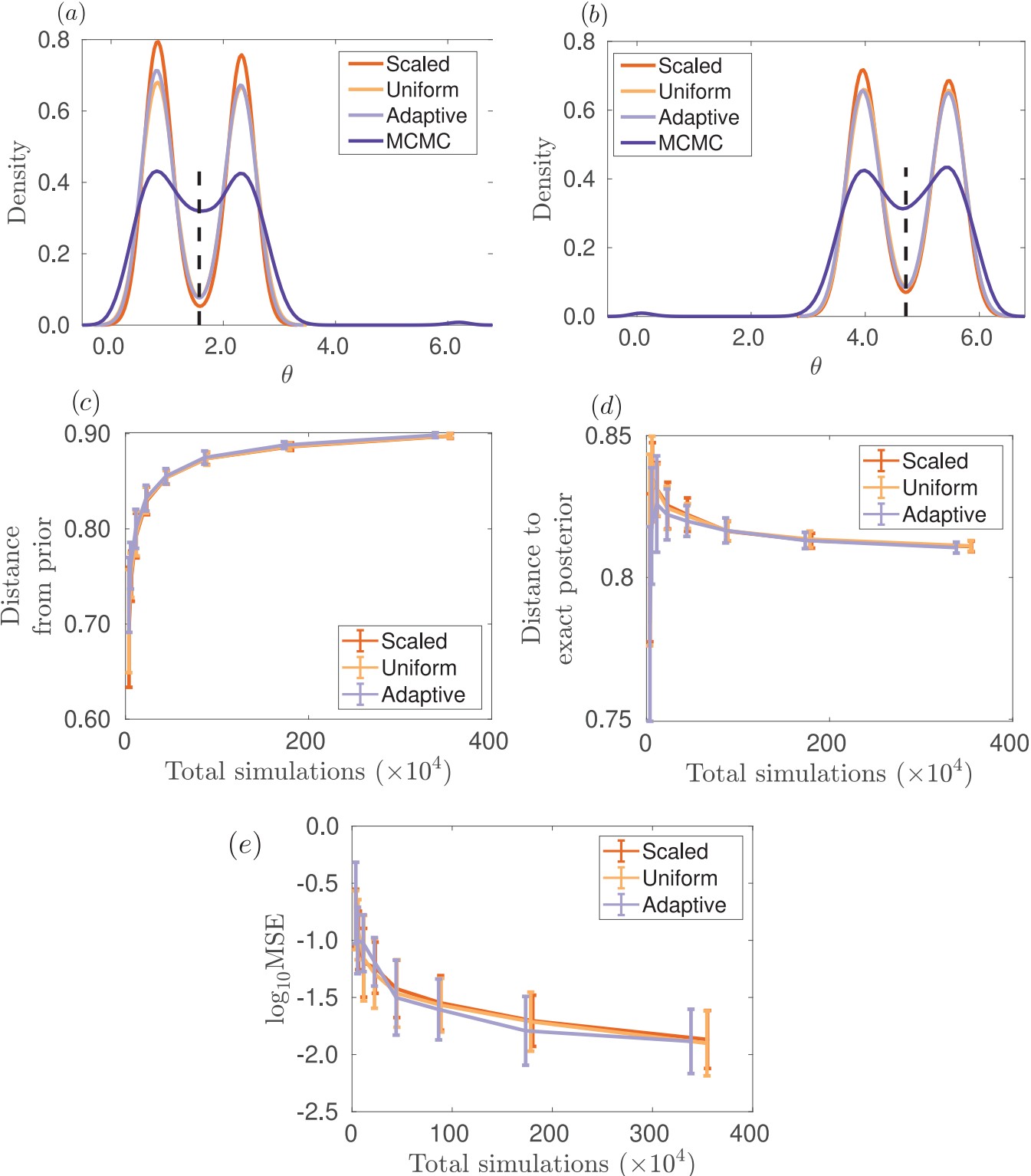

**Fig 2. Posterior for parameters θ of the bimodal toy model for different weights in the ABC distance function.** The posterior distribution for parameters θ = (θ₁, θ₂) of the bimodal toy model for different weights in the ABC distance function is shown in (a) and (b). ABC-SMC was used to provide estimates of the posterior, with T = 10 generations and N = 2, 000 particles at each generation with the posterior constructed from the closest 50% of the simulations (α = 0.5). Metrics to evaluate the performance of Algorithm 2 are shown in (c), (d), and (e) as N varies resulting in different total numbers of simulations from the model. Results are averaged over 40 repeated runs. In (c), is shown the Hellinger distance from the prior to approximate posterior distribution. In (d), the

Hellinger distance from the approximate posterior to the exact posterior distribution is displayed. In (e), the MSE from the approximate posterior mean to the point $(\pi/2, 3\pi/2)$ is shown. The orange, yellow and light purple lines show the scaled, uniform and adaptive methods, respectively. For comparison as a gold standard for this problem, the dark purple line shows the posterior obtained with MCMC using the exact likelihood without any ABC approximation. The true parameter value used to simulate the observed data is indicated by the vertical dashed black line.

We take initial conditions $S_1(0) = 10^5$, $S_2(0) = 0$, $S_3(0) = 0$ and consider an observational time period of [0, 100] with $n = 8$ geometrically spaced observations (to capture the multiple timescales present), without observational noise. For the dimerization system, we take the time series $s(\mathbf{x}) = [S_1(t_0), \ldots, S_1(t_n), S_2(t_0), \ldots, S_2(t_n), S_3(t_0), \ldots, S_3(t_n)]$ as summary statistics and infer the four-dimensional parameter $\theta = (k_1, k_2, k_3, k_4)$. We note that for a choice of parameter $\theta^* = (1, 0.04, 0.002, 0.5)$, and the given initial conditions, we obtain a fast decay of species $S_1$ and accumulation of species $S_2$, followed by a slower decay of $S_2$ and accumulation of $S_3$ (see Fig 3(a)).

The results of parameter inference for this system can be seen in Fig 3. The true parameters used are $\theta = (1, 0.04, 0.002, 0.5)$, and we apply a prior uniform on the logarithm of the parameters over the intervals $[10^{-2}, 10^2]$, $[10^{-3}, 10^1]$, $[10^{-5}, 10^{-1}]$, $[10^{-3}, 10^1]$, respectively, for each parameter. Parameters $k_1$ and $k_2$ are clearly identified by the adaptive choice of weights. The fast transient behaviour initially involves reactions at rate $k_1$, while $k_2$ corresponds to the longer timescale accumulation of species $S_3$. Parameters $k_3$ and $k_4$ are harder to identify with broader resulting posteriors, but again the adaptive algorithm does a better job at excluding regions of search space than a uniform choice of weights, or a scaling with the standard deviation.

## 4.4 Simple spatial model

Spatial models produce very high dimensional data, containing information about dynamics in both space and time. Here, we consider a simple spatial model in one dimension to describe the spreading of particles by diffusion without volume exclusion. We divide our spatial domain $X \in [-1, 1]$ into $m$ boxes or voxels, and label the numbers of particles in voxels 1, ..., $m$ as $S_1, \ldots, S_m$, respectively. Particles can jump between neighbouring voxels at rate $\theta = D/h^2$, where $D$ is the macroscopic diffusion constant and $h$ is the width of the voxel. We assume zero flux conditions at $X = \pm 1$ and take $m = 8$, so that $h = 1/4$. As an initial condition, we place 10 particles in each of the $m/2$ voxels on the left-hand side of the domain where $x < 0$, and allow the system to evolve over the time interval [0, 20]. We observe the system at $n = 8$ equally spaced time points, and take as our summary statistic the time series for each voxel, $s(\mathbf{x}) = [S_1(t_0), \ldots, S_1(t_n), S_2(t_0), \ldots, S_2(t_n), S_m(t_0), \ldots, S_m(t_n)]$, where $S_i(t_j)$ is the number of particles in voxel $i$ at time point $t_j$. Using synthetic data simulated with $\theta = 0.1$, we attempt to recover the jump rate $\theta$. The results of parameter inference for this problem are shown in Fig 4, where we have used a prior uniform on $\log_{10}(\theta)$ over the interval $[10^{-4}, 10^0]$. We successfully obtain an informative unbiased posterior for $\theta$ using the adaptive choice of weights, with a notable improvement in comparison to the other methods for selecting the weights.

## 4.5 Comparison of computational costs

If our proposed approach of adapting the weights of each of the summary statistics is to be used in practice, we must ensure that the increases in the quality of the resulting posterior justify the computational overhead required for optimizing the weights of the ABC distance function. Otherwise, it would be preferable simply to generate the posterior using ABC-SMC with more samples, but with fixed summary statistic weights. Therefore we are interested in

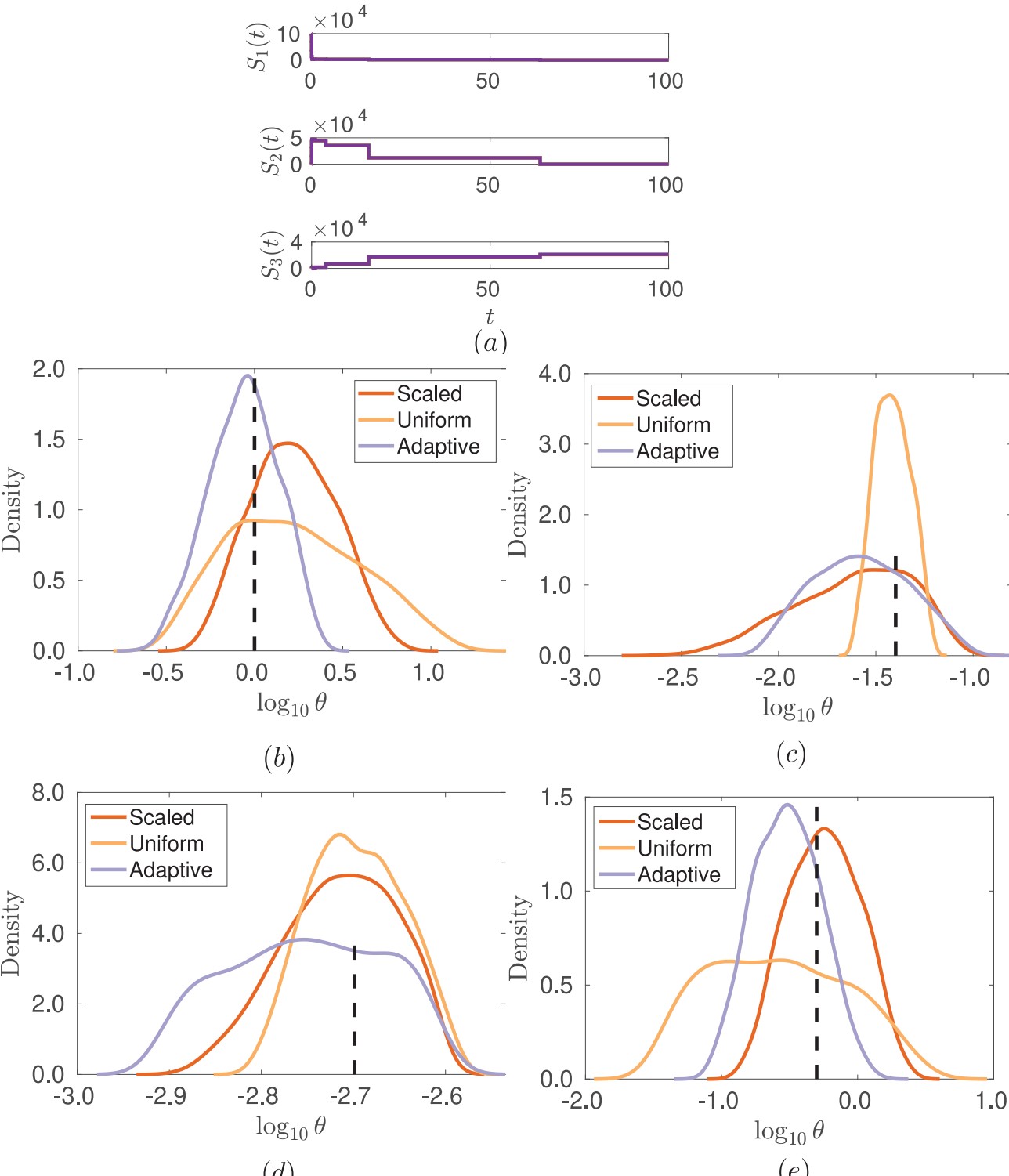

**Fig 3. Posteriors for parameters $\theta$ of the dimerization system for different weights in the ABC distance function.** Posteriors for parameters $\theta = (k_1, k_2, k_3, k_4)$ in the dimerization system for different weights in the ABC distance function. ABC-SMC was used with 10 generations and $N = 2000$ particles at each generation with the posterior constructed from the closest 50% of the simulations ($\alpha = 0.5$). (a) shows typical output from the model for the true parameters, for each species, $S_i$. Posterior marginal distributions for parameters $k_1, k_2, k_3, k_4$ are shown in (b) to (e). The orange, yellow and light purple lines show the scaled, uniform and adaptive methods, respectively. The true parameter values used to simulate the observed data is indicated by the vertical dashed black line.

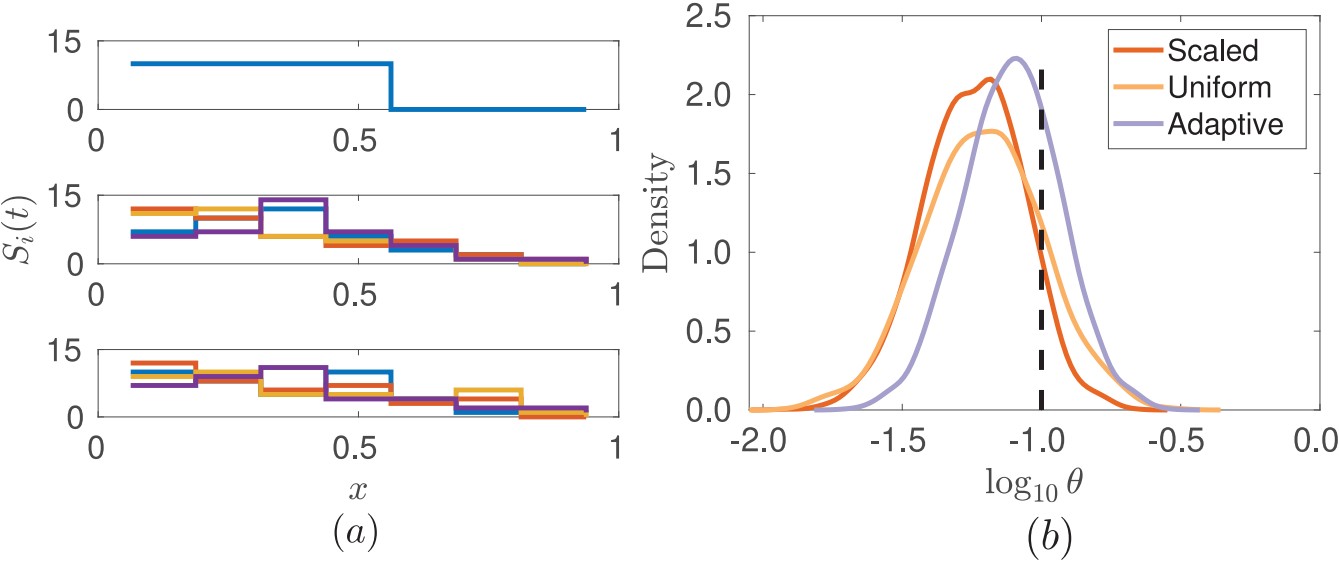

**Fig 4. Posteriors for parameter $\theta$ in the simplediffusion model for different weights in the ABC distance function.** Posteriors for parameter $\theta$ in the simple diffusion model for different weights in the ABC distance function. ABC-SMC was used for the inference with 10 generations and $N = 2000$ particles at each generation with the posterior constructed from the closest 50% of the simulations ($\alpha = 0.5$). (a) shows the spatial profile at three different time points ($t = 0$, 10, 20) and demonstrates the variability in the output for this spatial process across four realizations with the same parameter, $\theta = 0.1$. In (b), we compare the posteriors obtained for $\theta$ with different choices of weights. The orange, yellow and light purple lines show the scaled, uniform and adaptive methods, respectively. The true parameter value used to simulate the observed data is indicated by the vertical dashed black line.

evaluating the computational overheads of the search process, and how to limit the cost of the search in higher dimensions.

For each of the test problems described in Section 4, we compare a baseline scenario using ABC-SMC with uniform weights, with a second scenario where optimization of summary statistic weights is performed as in Algorithm 2, and with a third scenario such that an equivalent amount of additional computational effort is devoted to further model simulations so that more particles can be used in ABC-SMC with uniform summary statistic weights. Using the dimerization test problem, as described in Section 4.3, we ran Algorithm 2 with $N = 5,000$, $\alpha = 5\%$. To compare this to ABC-SMC with uniform weights, we performed parameter inference with uniform weights using both $N_1 = 5,000$, $\alpha_1 = 5\%$ and $N_2 = 5,600$, $\alpha_2 = 4.46\%$. The value of $N_2$ was chosen such that an equal length of computation time was spent in the search steps to find the summary statistic weights in Algorithm 2, as was spent in generating extra samples in ABC-SMC with uniform weights. A corresponding lower value of $\alpha$ was chosen so that the number of particles in the parameter sample was equivalent. Due to the differing computational complexities of each model, different values of $N_2$ and $\alpha_2$ will be used for each test problem.

In this case, adaptively choosing weights using Algorithm 2 resulted in a significantly greater distance between the prior and posterior, and reduced the bias in the posterior compared to running ABC-SMC with more samples, as measured by the distance between the maximum posterior estimate and the true parameters. These results, which represent improvements in the posterior for the same computational cost, are shown in Table 1 and the same procedure was used for the other test problems.

## 4.6 Consistent weights

Ideally, our search process should find the global optimum weight vector, so that if Algorithm 2 is run multiple times the same weight vector is obtained. In practice, for the examples we

**Table 1. Performance of Algorithm 2 compared with increasing the number of samples in ABC-SMC.** Results are shown for each of the test problems in the form: ABC-SMC with $N_1$ and $\alpha_1$ / ABC-SMC with $N_2$ and $\alpha_2$ / Algorithm 2 with $N$ and $\alpha$. Highlighted in bold is the method with best performance according to each metric.

| Test problem | Hellinger distance between prior and posterior | Bias in posterior |
|---|---|---|
| Uniform toy model | 0.7998 / 0.8088/ **0.8275** | 0.184 / 0.141 / **0.134** |
| Bimodal toy model | 0.865 / **0.867** / 0.866 | **0.029** / 0.034 / 0.031 |
| Dimerization | 0.921 / 0.919 / **0.921** | 0.563 / 0.493 / **0.421** |
| Diffusion | 0.700 / 0.710 / **0.743** | 0.533 / 0.475 / **0.125** |

have explored, the function to be optimized (distance between prior and posterior as a function of the distance weights) is very flat with respect to some of the distance weights. This makes it hard to consistently identify a global maximum. In Fig 5, we explore how the chosen weights vary for the uniform toy model. We can interpret this as the algorithm identifying the informative summary statistics and appropriately using the information from these, while allowing weights for other summary statistics to take a range of values without much effect on the resulting posterior. The largest weight is given to the most informative summary statistic. In the case of the uniform toy model, this is $s(\mathbf{x}) = \max_i x_i$.

## 4.7 Comparison to dimensionality reduction methods

Adaptively choosing the summary statistic weights within the ABC distance function can be seen as achieving a similar goal to summary statistic dimension reduction techniques [33, 37]. These techniques either project high-dimensional summary statistics into a lower dimensional subspace, or select an optimal subset of summary statistics via some optimality criterion. In contrast, a similar effect is achieved here when the statistics are combined in the weighted Euclidean distance function, $d_\mathbf{w}(\mathbf{x_1}, \mathbf{x_2})$, by weighting summary statistics to take account of both their inherent scale, and also their relative contribution towards the posterior distribution.

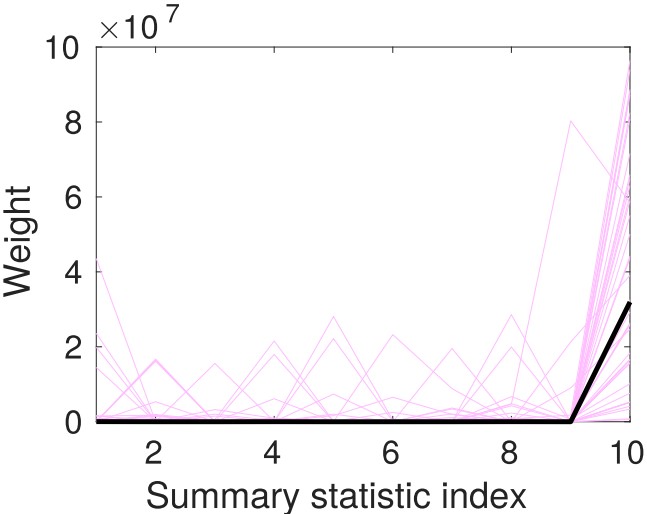

**Fig 5. The optimal distance weights found from the search procedure after 40 successive runs of Algorithm 2 on the toy model test problem (described in Section 4.1).** Parameters used are as for Fig 1. The faint purple lines show the resulting summary statistics weights from repeated runs of the ABC distance weight algorithm, while the black line shows the mean of the weights selected.

**Table 2. Comparison of the quality of the posteriors obtained using different methods to combine summary statistics.** Results given as adaptive method / Barnes et al. [35] / Fearnhead and Prangle [34]. Bold text highlights the best performance on a metric for a test problem.

| Test problem | Hellinger distance between prior and posterior | Bias in posterior |
|---|---|---|
| Uniform toy model | **0.822** / 0.806 / 0.815 | **0.055** / 0.059 / 0.056 |
| Bimodal toy model | **0.873** / 0.872 / 0.855 | 0.025 / **0.024** / 0.028 |
| Dimerization | **0.930** / 0.914 / 0.925 | 1.109 / 1.274 / **0.783** |
| Diffusion | **0.723** / 0.680 / 0.709 | **0.358** / 0.399 / 0.373 |

Uninformative summary statistics are automatically assigned a lower weighting, while more informative summary statistics are given high weights relative to their scale.

We tested our adaptive weight selection algorithm against the semi-automatic ABC method [34] (using the implementation available via abctools [52]), and the subset selection method of Barnes et al. [35] based on an approximate sufficiency criterion. In general, for the test problems considered, as described in Section 4, our method outperforms the competing methods, as shown by the metrics in Table 2. A larger value of the Hellinger distance indicates a greater distance between prior and posterior. The bias gives the distance between the posterior mean and the true parameter value. Values shown have been averaged over 10 successive runs of each method on different datasets. In implementing these methods, we have used only ABC rejection sampling, equivalent to a single generation of ABC-SMC, to compare the methods. This hinders our method which makes use of adaptive selection of weights, but is evaluated only on a single generation. In practice, these results mean that our method outlined in Algorithm 2 for adaptively choosing the weights of summary statistics produces a more informative posterior than competing methods based on dimensionality reduction of summary statistics.

## 5 Discussion

In this work, we have presented a method for improving the quality of posteriors resulting from approximate inference using ABC-SMC by optimizing the weights of the ABC distance function, $d_{\mathbf{w}}(s(\mathbf{x}_1), s(\mathbf{x}_2))$. By applying the methodology to several test problems we have demonstrated that our novel, adaptive method allows effective combination of summary statistics. We see superior performance using our algorithm in comparison with naive choices of uniform weights or using the scale of the summary statistics. Further benefits of adapting the weights include removing the requirement for design and selection of summary statistics 'by hand'.

### 5.1 Further work

Our method for automatically adapting the weights of the ABC distance function could be combined with other methods for dimensionality reduction of summary statistics to further improve the quality of posteriors produced with ABC for given computational effort. A particular area to consider would be how best to combine optimization of the distance weights for ABC and dimensionality reduction of the summary statistics. These are related approaches that can work well together. One approach that could be explored, for example, is enforcing some sparsity of the weights during the search step of the weights optimization. By setting some weights to be explicitly zero, we exclude the corresponding summary statistics, effectively reducing the dimensionality of our summary statistics. Further investigations could explore how best to sample sparse subsets of weights in high dimensions.

## 5.2 Conclusion

In summary, we propose a computationally efficient search procedure to identify a set of optimum weights to allow us to combine summary statistics within the ABC distance function in such a way that the gain in information in the posterior over the prior is maximized.

## Supporting information

**S1 File.**
(PDF)

## Author Contributions

**Conceptualization:** Jonathan U. Harrison, Ruth E. Baker.

**Investigation:** Jonathan U. Harrison.

**Methodology:** Jonathan U. Harrison.

**Software:** Jonathan U. Harrison.

**Supervision:** Ruth E. Baker.

**Validation:** Jonathan U. Harrison.

**Visualization:** Jonathan U. Harrison.

**Writing – original draft:** Jonathan U. Harrison.

**Writing – review & editing:** Jonathan U. Harrison, Ruth E. Baker.

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
