## [Decision Letter · Decision Letter 0]

28 Apr 2020

PONE-D-20-06109

An automatic adaptive method to combine summary statistics in approximate Bayesian computation

PLOS ONE

Dear Mr Harrison,

Thank you for submitting your manuscript to PLOS ONE. After careful consideration, we feel that it has merit but does not fully meet PLOS ONE’s publication criteria as it currently stands. Therefore, we invite you to submit a revised version of the manuscript that addresses the points raised during the review process.

As mentioned by the two referees the topic of the paper is relevant and the contributions can be of interesting. However, the manuscript needs considerable revision, including comparison with other methods and a number of important clarifications.

We would appreciate receiving your revised manuscript by Jun 12 2020 11:59PM. To enhance the reproducibility of your results, we recommend that if applicable you deposit your laboratory protocols in protocols.io, where a protocol can be assigned its own identifier (DOI) such that it can be cited independently in the future. For instructions see: http://journals.plos.org/plosone/s/submission-guidelines#loc-laboratory-protocols

We look forward to receiving your revised manuscript.

Kind regards,

Inés P. Mariño, Ph.D.

Academic Editor

PLOS ONE

Journal Requirements:

Reviewers' comments:

Reviewer's Responses to Questions

**Comments to the Author**

1. Is the manuscript technically sound, and do the data support the conclusions?

Reviewer #1: Partly

Reviewer #2: Yes

2. Has the statistical analysis been performed appropriately and rigorously? 

Reviewer #1: No

Reviewer #2: Yes

3. Have the authors made all data underlying the findings in their manuscript fully available?

Reviewer #1: Yes

Reviewer #2: Yes

4. Is the manuscript presented in an intelligible fashion and written in standard English?

Reviewer #1: Yes

Reviewer #2: Yes

5. Review Comments to the Author

Reviewer #1: This paper need a revision on presentation as the true posterior when available should be presented. Limit of applicability should be clearly stated, for instance, does it work with improper priors ?

Reviewer #2: The paper tackles a very interesting problem in the Bayesian context (within the intractable likelihood framework). The authors propose an adaptive procedure for selecting the summary statistic (and/or the distance, see my questions below) in an ABC method. The novel method is based on the nearest neighbour (NN) approach. As I said above, this work contains interesting material in my opinion. Moreover, the paper is well-written and well-structured. However, I have some suggestions for improving the quality of the manuscript and its possible impact. See below.

My first two observations are even more general than the contents and goals of the paper (they are regarding ABC techniques in general).

- It would be very interesting if the authors clarify (in the introduction of Section 1.2 with a clear remark) what is the difference in choosing a distance and a summary statistics in ABC, or directly assume an approximation of the likelihood function (and then use standard inference computational methods). To be more clear: if in your algorithm 1 (or just an ABC-rejection method) you use a soft condition considering for instance a weighting function \\exp(-d(s(x),s(y))) (instead of an hard condition as you have in line 10 of Algorithm 1), I believe that the ABC computational methods (ABC-SMC or ABC-rejection, etc.) can be interpreted as standard computational techniques assuming a particular approximation of the likelihood. Then, choosing the distance and summary statistic is equivalent to approximate your “costly” and/or intractable likelihood function.

I believe that this discussion/clarification is relevant for your paper.

- Another point to clarify: you have the distance d(.,.) and the summary statistic s(x), but actually we need to learn the combination of both d(s(x),s(y)). What you are “learning”? d(.,.), s(x) or both (in the sense of “both together ”)? I believe the last option. But it should be clarified in different part of the text (also in the abstract and introduction).

For instance, I believe that the method in M U Gutmann, R Dutta, S Kaski, and J Corander. Likelihood-free inference via classification. Statistics and Computing,8(2):411-425, 2018, is learning both together of just the distance d(.,.)? please clarify also this point, discussing some connections to the other papers in the literature.

- Related to the previous point: the first formula in Section 2 seems to show that you are only learning the distance (or more specifically, the weights in a weighted Euclidean distance). Please clarify.

- It is not clear why you choose the Hellinger distance. Is there some specific reason? Moreover it seems that you use a more generic distance at page 6. Please clarify.

- Regarding the state-of-the-art in the introduction: it is quite poor. In order to improve the quality of the paper and its impact, I suggest to extend the state-of-the-art discussion including also references to noisy Monte Carlo methods and other computational methods for intractable likelihood, for instance,

Murray, I., Ghahramani, Z., MacKay, D.: MCMC for doubly-intractable distributions. Proceedings of the Twenty-Second Conference on Uncertainty in Artificial Intelligence (UAI2006), 1-8 (2006)

Moller, J., Pettit, A., Reeves, R., Bertheksen, K.: An efficient Markov Chain Monte Carlo method for distributions with intractable normalising constants. Biometrika 93, 451-458 (2006)

Alquier, P., Friel, N., Everitt, R., Boland, A.: Noisy Monte Carlo: Convergence of Markov chains with approximate transition kernels. Statistics and Computing 26(2), 29-47 (2016)

Medina-Aguayo, F.J., Lee, A., Roberts, G.O.: Stability of noisy Metropolis-Hastings. Statistics and Computing 26(6), 1187-1211 (2016)

The particle MCMC methods could be included within the the pseudo-marginal approaches (see your ref. [1]) and also with the multiple Try Metropolis (MTM) algorithms. Indeed, evening the MTM schemes we can consider that there is an implicit approximation of the marginal likelihood seem for instance,

L. Martino, "A Review of Multiple Try MCMC algorithms for Signal Processing", Digital Signal Processing, Volume 75, Pages: 134-152, 2018

L. Martino, F. Leisen, J. Corander, "On Multiple Try Schemes and the Particle Metropolis-Hastings Algorithm", viXra:1409.0051, 2014.   

I believe that this point also deserves to be mentioned.

- Can you say more regarding the adaptation of the tolerance in Algorithm 2? this is quite important since the distance and the tolerance plays an complementary role in your hard condition at line 19 of Algorithm 2. How robust is your algorithm with respect to a change of the adaptation of the tolerance? can you show some results keeping fixed the tolerance?

- Your algorithm 2 has the additional computational cost of optimizing your weights. You should compare with a standard ABC algorithm without optimization but with extra- samples (i.e., with more samples than your method) in order to have a fair comparison with your method.

6. PLOS authors have the option to publish the peer review history of their article (what does this mean?). If published, this will include your full peer review and any attached files.

Reviewer #1: No

Reviewer #2: No

---

## [Author Response · Author response to Decision Letter 0]

14 Jul 2020

Response to Referee 1

We would like to thank Referee 1 for taking the time to consider our manuscript, and provide constructive comments for its improvement.

The methods does not apply when priors are improper priors. Although this is not usual in ABC, in the end we don’t know what happen if priors are very spread out on the support of parameter space. For instance, would this imply uniform weights?

In the case of improper priors, the Hellinger distance from prior to approximate posterior is infinite irre- spective of the choice of weights. The method of Algorithm 2 cannot be applied in this case. We would instead recommend using a weakly informative prior distribution (see Gelman et al. [2014]) that contains enough information to regularise the posterior distribution without capturing full scientific consensus knowledge about a parameter.

The case of proper priors that are very spread out has been considered in this work. For the uniform toy model, the dimerization system and the simple spatial model in Sections 4.1, 4.3 and 4.4, we used priors uniform on the logarithm of the parameters θ, which is a diffuse non-informative choice of prior distribution. Provided the Hellinger distance from prior to approximate posterior is finite, and varies when the ABC distance weights are adjusted such that a maximum can be found, then the method of Algorithm 2 should have the correct behaviour. There is no reason why uniform weights within the ABC distance function should be optimal when the prior is very spread out.

To demonstrate this, we have added Section C and Figure S4 to the Appendix. In this Section, we show, for a version of the Gaussian toy problem, that the adaptive method of Algorithm 2 is able to offer an improved approximation to the true posterior compared to a uniform choice of weights in the ABC distance function regardless of increasing variance in the prior distribution.

All comparison with the methods I think are misleading, in fact the obtained posteriors are compared among them but not with the true posterior in the case when this is known by using analytical likelihood. For example, in the case of Uniform or Normal example I would have drawn the true posterior instead of the true value of the parameter which is just a point.

We thank the reviewer for highlighting this point. We have made comparison with the exact MCMC approximation to the posterior based on the analytical likelihood for these cases. These are shown as the dark purple curves labelled as MCMC. We have removed the label for the point value used to simulate the dataset, and describe the dashed line indicating this point value in a caption only, which should avoid confusion on this point. The captions now read: "For comparison as a gold standard for this problem, the dark purple line shows the posterior obtained with MCMC using the exact likelihood without any ABC approximation. The true parameter value used to simulate the observed data is indicated by the vertical dashed black line."

Comparison should be done also in terms of computational effort at least in terms of the number of simulated data from the model. For instance, the uniform assignment of weights maybe is not optimal, but it is certainly less computational demanding.

To address this point raised by both reviewers, we have updated Section 4.5 and renamed this as "Comparison of computational costs". In Section 4.5, we fix a computational budget, and compare a situation where additional computational resources are allocated to optimization of the weights of the ABC distance function for Algorithm 2, versus a situation where additional computational resources are allocated to extra model simulations with uniform weights. In most cases of the models considered, it was more beneficial to optimize the weights of the ABC distance function, rather than to allocate resources to additional simulations. In addition, we have updated Section 4.5 such that we state the following: "If our proposed approach of adapting the weights of each of the summary statistics is to be used in practice, we must ensure that the increases in the quality of the resulting posterior justify the computational overhead required for optimizing the weights of the ABC distance function. Otherwise, it would be preferable simply to generate the posterior using ABC-SMC with more samples, but with fixed summary statistic weights." We further describe the implementation of this comparison in the text: "For each of the test problems described in Section 4, we compare a baseline scenario using ABC-SMC with uniform weights, with a second scenario where optimization of summary statistic weights is performed as in Algorithm 2, with a third scenario such that an equivalent amount of additional computational effort is devoted to further model simulations so that more particles can be used in ABC-SMC with uniform summary statistic weights." Results evaluating the posterior approximations produced in each case are compared in Table 1.

Page 3 lines 72-75 these could be rephrased.

We thank the reviewer for this suggestion, and have rephrased these lines to read: “Fearnhead and Prangle [13] developed a popular method to find informative linear combinations of summary statistics by fitting a regression for each model parameter.”

Response to Referee 2

We would like to express our gratitude to Referee 2 for taking the time to consider our manuscript, and provide constructive comments for its improvement.

It would be very interesting if the authors clarify (in the introduction of Section 1.2 with a clear remark) what is the difference in choosing a distance and a summary statistics in ABC, or directly assume an approximation of the likelihood function (and then use standard inference computational methods). To be more clear: if in your algorithm 1 (or just an ABC-rejection method) you use a soft condition considering for instance a weighting function exp(−d(s(x),s(y))) (instead of an hard condition as you have in line 10 of Algorithm 1), I believe that the ABC computational methods (ABC-SMC or ABC-rejection, etc.) can be interpreted as standard computational techniques assuming a particular approximation of the likelihood. Then, choosing the distance and summary statistic is equivalent to approximate your ‘costly’ and/or intractable likelihood function.

We thank the reviewer for suggesting this perspective, and have added the following to the introduction in Section 1.1: "ABC can be viewed as providing a regular Bayesian analysis, but with an approximation to the likelihood function of the form [Sisson et al., 2018]:

pABC(y|θ) = � 1 [d(x, y) ≤ ε] p(y|θ)dx, (1)

where pABC(y|θ) is the ABC approximation of the likelihood, and p(y|θ) is the exact likelihood. The quality of the approximation depends on the choice of tolerance, ε, and distance function d(x, y)."

To further build on this interpretation, we now state at the end of Section 1.2: "Following the perspective of ABC offering an approximation to the likelihood, as in eq. (1), we can consider how optimizing the choice of ABC distance function, given a set of summary statistics, can improve the approximation to the likelihood offered by ABC."

Another point to clarify: you have the distance d(., .) and the summary statistic s(x), but actually we need to learn the combination of both d(s(x), s(y)). What you are ‘learning’? d(., .), s(x) or both (in the sense of “both together ”)? I believe the last option. But it should be clarified in different part of the text (also in the abstract and introduction).

We appreciate the opportunity to clarify this. We assume that a list of possible summary statistics is available, which is a common assumption and broadly applicable given some domain expertise about an application. Rather than selecting which of these summary statistics to include in the analysis, we use all the summary statistics, and select a distance function, such that changes in irrelevant summary statistics can be ignored, and informative statistics considered on equivalent scales. We are learning only d(. . . ) and not s(x).

In the abstract, we state that in this work we are "developing an automatic, adaptive algorithm that chooses weights for each summary statistic" and that we are "adapting the weights within the ABC distance function". In the text in Section 1.2, we have now updated our wording to emphasise this as follows: "In this work, we approach the problem from the point of view of finding the right distance func- tion, adapted to information contained in a fixed list of known summary statistics, rather than selecting a certain subset of summary statistics."

For instance, I believe that the method in M U Gutmann, R Dutta, S Kaski, and J Corander. Likelihood-free inference via classification. Statistics and Computing,8(2):411-425, 2018, is learning both together of just the distance d(.,.)? please clarify also this point, discussing some connections to the other papers in the literature.

We have added additional discussion and highlighted connections to other work in this area as follows: "Other work has avoided using summary statistics at all by considering Wasserstein distance between full data sets [Bernton et al., 2017, 2019] or by using classification techniques from machine learning to discriminate between datasets [Gutmann et al., 2018], and thus choosing a classification method and features rather than a distance function and summary statistics."

Related to the previous point: the first formula in Section 2 seems to show that you are only learning the distance (or more specifically, the weights in a weighted Euclidean distance). Please clarify.

As discussed following the previous comment, in this work we are learning the distance and not the summary statistics.

It is not clear why you choose the Hellinger distance. Is there some specific reason? Moreover it seems that you use a more generic distance at page 6. Please clarify.

We thank the reviewer for raising the choice of distance function. Alternative choices of distance func- tion could be used in a similar approach. We found that the Hellinger distance was more robust to distributions with different support than other distances such as the Kullback-Leiber (KL) divergence. In the text we address this issue as follows: "The Hellinger distance is finite when comparing distributions with different support (unlike the KL divergence). This property is desirable when comparing a broad prior with a posterior distribution where we have gained some knowledge of parameter space and can exclude certain regions."

The nearest neighbour estimator described on page 6 is valid for a wider class of α-divergences, Dα, than the Hellinger distance and we describe the methods in this context. Equation (2) on page 6 is given for this more general case of α-divergences, Dα, rather than specifically for the Hellinger distance. This offers consistency with notation in previous literature.

Regarding the state-of-the-art in the introduction: it is quite poor. In order to improve the quality of the paper and its impact, I suggest to extend the state-of-the-art discussion including also references to noisy Monte Carlo methods and other computational methods for intractable likelihood.

We have significantly extended the discussion of the state-of-the-art in the introduction in Section 1 beyond ABC methods and have included references to noisy Monte Carlo and other intractable likelihood techniques.

Can you say more regarding the adaptation of the tolerance in Algorithm 2? this is quite important since the distance and the tolerance plays an complementary role in your hard condition at line 19 of Algorithm 2. How robust is your algorithm with respect to a change of the adaptation of the tolerance? can you show some results keeping fixed the tolerance?

One of the main benefits of using the quantile α of the distribution of distances to help set the tolerance schedule is that it avoids needing to consider directly the scale of typical distances between observed and simulated data. When optimizing the distance function, we cannot be sure what the scale of typical distances between observed and simulated data will be, as these depend of the summary statistic weights. This makes it harder to determine a fixed schedule of tolerances in advance.

Nonetheless, we have added Section D in the Appendix and Figure S5, where we provide results with a fixed tolerance schedule determined in advance, and thus demonstrate robustness to changes in the adaption of the tolerance. In the text though, we stress that "we recommend using a tolerance determined as a quantile of the distance distribution, as this avoids the need to set a fixed tolerance schedule in advance when the scale of typical distances between observed and simulated data is not known in advance. A fixed tolerance scheme prevents fair comparison between uniform, scaled and adaptive weighting methods for the weights of the ABC distance function, as different numbers of model simulations are required in each case dependent on the tolerance schedule. By using the quantile α of the distribution of distances, we ensure that an equal number of model simulations are used in each case. Methods to design tolerance schedules have been investigated by Silk et al. [2013] based on a threshold-acceptance rate curve and could be considered as an alternative."

Your algorithm 2 has the additional computational cost of optimizing your weights. You should compare with a standard ABC algorithm without optimization but with extra- samples (i.e., with more samples than your method) in order to have a fair comparison with your method.

We thank the reviewer for also raising the issue of computational cost. We now perform this comparison in Section 4.5 and have made significant updates to this Section, including renaming it as "Comparison of computational costs" for additional clarity. Further details are described above in the response to Referee 1.

References

A Gelman, J B Carlin, H S Stern, D B Dunson, A Vehtari, and D B Rubin. Bayesian Data Analysis, volume 2. CRC press Boca Raton, FL, 2014.

S A Sisson, Y Fan, and M Beaumont. Handbook of approximate Bayesian computation. CRC Press, 2018.

E Bernton, P E Jacob, M Gerber, and C P Robert. Inference in generative models using the Wasserstein distance. arXiv preprint arXiv:1701.05146, 2017.

E Bernton, P E Jacob, M Gerber, and C P Robert. Approximate bayesian computation with the Wasser- stein distance. Journal of the Royal Statistical Society: Series B (Statistical Methodology), 81(2): 235–269, 2019.

M U Gutmann, R Dutta, S Kaski, and J Corander. Likelihood-free inference via classification. Statistics and Computing, 28(2):411–425, 2018.

D Silk, S Filippi, and M P H Stumpf. Optimizing threshold-schedules for sequential approximate bayesian computation: applications to molecular systems. Statistical Applications in Genetics and Molecular Biology, 12(5):603–618, 2013.

---

## [Decision Letter · Decision Letter 1]

17 Jul 2020

An automatic adaptive method to combine summary statistics in approximate Bayesian computation

PONE-D-20-06109R1

Dear Dr. Harrison,

We’re pleased to inform you that your manuscript has been judged scientifically suitable for publication and will be formally accepted for publication once it meets all outstanding technical requirements.

Kind regards,

Inés P. Mariño, Ph.D.

Academic Editor

PLOS ONE

Additional Editor Comments (optional):

Reviewers' comments:

Reviewer's Responses to Questions

**Comments to the Author**

1. If the authors have adequately addressed your comments raised in a previous round of review and you feel that this manuscript is now acceptable for publication, you may indicate that here to bypass the “Comments to the Author” section, enter your conflict of interest statement in the “Confidential to Editor” section, and submit your "Accept" recommendation.

Reviewer #1: All comments have been addressed

Reviewer #2: All comments have been addressed

2. Is the manuscript technically sound, and do the data support the conclusions?

Reviewer #1: Yes

Reviewer #2: Yes

3. Has the statistical analysis been performed appropriately and rigorously? 

Reviewer #1: Yes

Reviewer #2: Yes

4. Have the authors made all data underlying the findings in their manuscript fully available?

Reviewer #1: Yes

Reviewer #2: No

5. Is the manuscript presented in an intelligible fashion and written in standard English?

Reviewer #1: Yes

Reviewer #2: Yes

6. Review Comments to the Author

Reviewer #1: The authors satisfactory answered to the points I raised.

Reviewer #2: (No Response)

7. PLOS authors have the option to publish the peer review history of their article (what does this mean?). If published, this will include your full peer review and any attached files.

Reviewer #1: No

Reviewer #2: No

---

## [Editor Report · Acceptance letter]

24 Jul 2020

PONE-D-20-06109R1 

An automatic adaptive method to combine summary statistics in approximate Bayesian computation 

Dear Dr. Harrison:

I'm pleased to inform you that your manuscript has been deemed suitable for publication in PLOS ONE. Congratulations! Your manuscript is now with our production department. 

Kind regards, 

on behalf of

Dr. Inés P. Mariño 

Academic Editor

PLOS ONE